# Digestion Profiles of Protein in Edible Pork By-Products

**DOI:** 10.3390/foods11203191

**Published:** 2022-10-13

**Authors:** Xianming Zeng, Bowen Lv, Kexin Zhang, Zhe Zhu, Qiuyue Li, Bulei Sheng, Di Zhao, Chunbao Li

**Affiliations:** 1School of Food Science and Technology, Nanjing Agricultural University, Nanjing 210095, China; 2School of Tea and Food Science & Technology, Hefei 230036, China

**Keywords:** pork by-products, digestibility, peptidomic analysis

## Abstract

Edible pork by-products are widely consumed in many areas, whereas their digestion characteristics have rarely been evaluated. This work compared the digestibility of protein in boiled pork liver, heart, tripe and skin with tenderloin as a control. Cooked skin showed the highest digestibility in the simulated gastric digestion, whereas its gastric digests were less digested in the simulated intestinal stage. In contrast, cooked tripe showed the lowest gastric digestibility but relatively higher intestinal digestibility. All the edible by-products showed lower digestibility than tenderloin, especially for pork liver, in which large undigested fractions (>300 μm) could be observed. Corresponding to these results, larger amount of bigger peptides was found in the digests of pork liver and skin. In addition, peptides in tripe (average bioactive probability = 0.385) and liver digests (average bioactive probability = 0.386) showed higher average bioactive probability than other samples. Tripe digests contained the highest level of free Asp, Gln, Cys, Val, Phe, Pro, Ser, Thr, Ile and Asn, whereas heart digests contained the highest level of free Leu, Met and Arg. These results could help to reveal the nutrition value of pork by-products.

## 1. Introduction

A large amount of by-products from animals (pig, beef, chicken and fish) are produced, which weigh around 20–50% of the relevant animals [1,2,3]. Taking China as an example, approximately 163 million pigs were slaughtered in 2020, producing over 6 million tons of pig blood, 60 million tons of pig bone and almost 1 million tons of viscera [1]. Livestock and poultry by-products approximately account for 10–15% value of animals in developed countries. Proper usage of these by-products significantly contributes to the profit of meat industry and helps to reduce the pollution induced by discarding. By-products of the livestock and poultry mainly consist of bone, blood and edible parts. Catgut and sausage casing can be produced from intestine of livestock. Collagen gelatin and elastin are the main targets extracted from skin, which are widely used as emulsifier and stabilizer of jelly, ice cream and yogurt [1,4,5]. Coenzyme A, heparin sodium and its hydrolysate are the main products extracted from the liver of livestock [1,6]. In addition, manufacture of peptides with antihypertensive, antioxidant, antimicrobial, immunomodulatory, anticholesterol, satiety, organoleptic and antidiabetic activity from bone and blood have also been widely reported [7,8,9,10]. Pancreatin, cholic acid, pepsin, thymosin and coenzyme Q can also be isolated from related pig by-products.

Edible by-products from animals, including liver, heart, stomach and skin, are widely consumed, especially in China, Japan, France, Germany and some southeast Asian countries. There are plenty of favorable cuisines involving in edible by-products from animal, including pork belly bag chicken, andouillette, and Palatinate sausage. These edible products were also commonly used in the preparation of luncheon meat pet food [1]. Intake of by-products can supplement trace elements and vitamins, whereas some assumed toxic minerals also need additional attention [11]. By-products usually have high contents of protein, which could be an important protein source in the future, considering the increasing population and demand for protein. These edible by-products also vary in protein compositions and muscle texture. For example, pork skin is rich in collagen and elastin, and the texture of smooth and cardiac muscles in pork tripe or heart are largely different from skeletal muscle. Therefore, the digestibility of proteins in these edible by-products should be largely different from that in traditional pork, which is closely related to their nutritional values. To verify this speculation, the digestibility of pork liver, heart, tripe and skin were compared with that of tenderloin, a type of widely-consumed pork by detecting the liberation of free amino group (–NH_2_), hydrolysis of proteins during in vitro digestion, as well as analyzing the free amino acids and peptides fingerprints in digests.

## 2. Materials and Methods

### 2.1. Materials

Tenderloin (n = 3), pork liver (n = 3), heart (n = 3), triple (n = 3) and skin (n = 3) were collected from local Huarun Suguo Supermarket (Nanjing, China), these sample was collected in ice box and transported to freezer house (−20 °C) within 1 h after collection. Pepsin (≥250 unit/mg) pancreatin (8 × USP), Fluorescamine (≥98%), leucine (≥98%), Nile Red (≥97%) and Nile Blue (≥95%) were purchased from Sigma-Aldrich (Shanghai, China).

### 2.2. Simulated Digestion

Before digestion, tenderloin and by-products were transferred from the −20 °C freezer house to a 4 °C cold room. After overnight unfreezing, tenderloin and each by-product were transferred into a vacuum bag and packed by a vacuum packaging machine (PROMAX, Promarksvac, CA, USA) and then heated at 90 °C for 1 h. The protein content in cooked tenderloin, liver, heart, tripe and skin was tested by Kjeldah method (n = 3) and was determined as 25.7%, 35.2%, 32.8%, 27.3% and 22.9%, respectively.

Simulated gastric fluid (SGF) and simulated intestinal fluid (SIF) were prepared according to Wang et al. [12]. Each sample (3.0 g) in pieces was added in 7 mL of SGF, and the mixture was homogenized at 9500 rpm for 40 s. The pH of the mixture was adjusted to pH 3.0 using 1 M hydrochloric acid. Then, 10 mg/mL of pepsin was added to obtain a final concentration of 500 units/mL to initiate the gastric digestion, which was conducted in a shaking bath at 37 °C and 200 rpm. After 15, 60 and 120 min of simulated gastric digestion, 1.5 mL of samples were obtained and mixed with 1.5 mL of SIF to elevate the pH to around 7.5 and stop the action of pepsin. The remaining gastric digests were mixed with the same amount of SIF. Pancreatin solution (2 mg/mL, 1 mL) was added to initiate the simulated intestinal digestion, which was conducted in a shaking bath at 37 °C and 200 rpm. After 15, 60 and 120 min of incubation, samples were collected and inactivated enzymes at 95 °C for 5 min. Digestion of each sample was repeated three times. All samples were stored at −20 °C before further analysis. The digestion of tenderloin and each kind of by-product was repeated 3 times.

### 2.3. Fluorescamine Assay

Liberation of –NH_2_ during simulated digestion was measured using a fluorescamine method [13]. Each digest (75 μL) was mixed with TCA (75 μL, 24%, *w*/*v*). The mixture was precipitated in ice bath for 30 min and was then centrifuged at 15,000× *g* and 4 °C for 20 min. The supernatant was collected and diluted using 1 mM HCl so that the read fell within the range of 0.05–3 mM standard L-leucine. Then, 30 μL of aliquot of digested samples, 300 μL of fluorescamine solution (0.2 mg/mL, dissolved in anhydrous acetone) and 900 μL of sodium tetraborate (0.1 M, pH 8.0) were mixed and transferred to Costar 96 Flat Black plate. Afterwards, fluorescent intensity was measured at a excitation wavelength of 390 nm and a emission wavelength of 480 nm in an Infinite M200 PRO microplate reader (Tecan, Grödig, Austria). The level of released –NH_2_ in each digest was determined according to the protein content as mentioned in Section 2.2 and the standard L-leucine curve. Measurement of each sample was repeated in triplicates.

### 2.4. SDS-PAGE

Each protein or digest was mixed with 5× sample buffer to obtain a concentration of 2 mg/mL, which was incubated in a water bath at 95 °C for 10 min to denature the proteins or digest. Subsequently, the samples (10 µL) were injected in a precast gel (4–20%), and electrophoresis was applied at 80 V for 30 min and at 120 V for another 90 min. Subsequently, the SDS-PAGE gel was stained, destained, and graphed.

### 2.5. Confocal Laser Scanning Microscopy (CLSM) Measurement

The digests were observed using CLSM method [14]. Then, 100 μL of gastric and intestinal digests was transferred in a 1.5 mL tube. Afterwards, 40 μL of 2% 1,2-propanediol aqueous solution containing 0.1% Nile Red and 1% Nile Blue fluorescent dye was added in the tube, and the dyeing lasted in dark for 5 min. Each sample was transferred to the glass slides and covered using a square glass. These samples were observed under a 40× objective lens in a SP8 X CLSM (TCS, Leica, Wetzlar, Germany), using 552 nm and 633 nm laser to excite Nile Red and Nile Blue, respectively. The scanning frequency was set as 200 Hz. Leica Application Suite X software was used to analyze images.

### 2.6. Peptidomics Analysis

Peptide sequences were measured using a peptidomic protocol [13]. Each digest was mixed with equal volume of formic acid aqueous solution (0.2%). Then, a 10 kDa spin filter was applied to filter the mixture at 14,000× *g* and 4 °C for 15 min. Then, a Zorbax 300SB-C18 peptide trap (Agilent Technology, Wilmington, DE, USA) connected with an RP-C18 separation column (0.15 mm × 150 mm, Column Technology) was used to separate the sample (10 μL). Programmed elution was applied using 0.1% formic acid as eluent A and 0.1% formic acid acetonitrile solution as eluent B, at a flow rate of 30 nL/min. The following gradient elution was processed: from 4% to 50% eluent B in the initial 50 min; from 50% to 100% eluent B from 50 to 54 min; and 100% eluent B from 54 to 60 min. The MS/MS data were collected in a Q Exactive (Thermo Scientific, Fremont, CA, USA) in a positive ion mode. The MS was scanned from 350 to 1800 *m*/*z*, and the MS/MS was scanned from 150 to 1800 *m*/*z*. The archived mass data were searched using a Proteome Discover 1.4 software (Thermo Fisher) with a false discovery rate of 1%, a tolerance of the peptide as 20 ppm, and the tolerance of MS/MS as 0.1 Da. Measurement of each sample was repeated twice, and the identical peptides observed in both measurements are accepted.

### 2.7. Determination of Free Amino Acid

Cooked sample (2 g) was finely chopped into small pieces (approximately 2 mm) and mixed in 10 mL 50% acetonitrile. Then, a centrifugation (13,000 r/min, 10min) was applied to obtain the supernatant, which was filtrated through the polytetrafluoroethylene microfiltration membrane (0.22 μm). Then, 2 µL of filtrate was injected into the TSQ Quantum Ultra system. MSLab HP-C18 (150 mm × 4.6 mm, 5 μm) was used for the separation of each sample. A programmed elution at a flow rate of 0.4 mL/min was applied, with eluent A containing 0.15% formic acid and 10 mmol/L ammonium formate, and eluent B containing 85% acetonitrile solution and 10 mmol/L ammonium formate. The detailed gradient elution process was conducted as follows: 0–6 min, 100% eluent B; 6–12 min, from 100% eluent B to 70% eluent B; 12–18 min, 70% eluent B; 18–18.5 min, from 70% to 100% eluent B; and 18.5–21 min, 100% eluent B. External standard was used for the quantitative analysis of each amino acid.

### 2.8. Statistics Analysis

The differences in liberated –NH_2_ (n = 3) and free amino acid (n = 3) between the samples were analyzed by one-way ANOVA (analysis of variance) under a Duncan’s multiple range test in SAS system V8. Differences were considered to be significant when the p value was smaller than 0.05.

## 3. Results and Discussion

### 3.1. Release of –NH_2_ Residues during Simulated Digestion

The liberation of –NH_2_ during simulated digestion was shown in Table 1, indicating the hydrolysis degree of peptide bond during in vitro digestion. In the initial 15 min of gastric digestion, skin protein was shown to be hydrolyzed to the greatest degree (released 3.3 μmol/g protein –NH_2_), which was followed by that in the digests of cooked tenderloin, liver, tripe and heart in a descending order. Therefore, skin protein should be the easiest to be digested in the initial gastric digestion. After 120 min of gastric digestion, 6.8 μmol/g protein of –NH_2_ was released in the digests of cooked skin, which was followed by cooked tenderloin (6.7 μmol/g protein), heart (3.7 μmol/g protein), liver (3.3 μmol/g protein) and tripe (1.9 μmol/g protein) in a descending order. These results demonstrated that skin proteins had the highest digestibility, whereas tripe, liver and heart proteins were more difficult to be digested during the simulated gastric digestion. The gastric mucosa structure remain in the tripe were resistant to the hydrolysis of pepsin and account for the low digestibility of tripe [15].

After entering the simulated intestinal digestion stage, digests of cooked tenderloin and heart further released 9.5 and 10.9 μmol/g protein of –NH_2_ in the first 15 min of intestinal digestion. In contrast, digests of triple, liver and skin digests released 5.5, 5.4 and 0.6 μmol/g protein of –NH_2_ in a descending order during the same period, demonstrating the lower digestion rate of them in the initial simulated intestinal digestion, especially for the skin sample. After the whole gastrointestinal digestion, all edible pork by-products showed significant lower digestibility than tenderloin, especially for liver and skin proteins. Cooked pork liver and skin only released 11.5 and 14.9 μmol/g protein of –NH_2_, which were much lower than that in cooked pork tenderloin (37.9 μmol/g protein) and heart (24.4 μmol/g protein). Collagen is a major protein in pork skin, which is rich in proline (Pro) and hydroxyproline [10]. The action of trypsin will be largely hindered by Pro, which could account for the low digestibility of skin in the simulated intestinal digestion [16]. In addition, different from the liver tissue of human, there are abundant connective tissue surrounding the hepatic lobule of the swine liver. Strong resistance of connective tissues to the action of proteases, which may account for the lower digestibility of cooked pork liver sample [17].

### 3.2. SDS-PAGE and CLSM Analysis of Digests

The SDS-PAGE (Figure 1) and CLSM (Figure 2) were further used to investigate the digestibility of by-products since undigested protein or big peptides could be observed. Along with the process of gastric digestion, proteins in all samples was gradually hydrolyzed, as shown in the SDS-PAGE. Aggregates with large molecular weight in cooked liver disappeared rapidly, whereas aggregates in other samples were only partially hydrolyzed. After the whole gastric digestion, longer skeletal muscle fiber and shorter cardiac muscle fibers could still be observed in the digests of tender lion and pork heart in CLSM images (Figure 2A). Large amount of amorphous aggregates was found in the gastric digests of pork tripe and liver, corresponding to their lower level of –NH_2_ (Table 1). Comparatively, dispersed digests were found in the gastric digests of pork skin. The smallest amount of undigested protein or big peptides was found in the gastric digest of cooked pork skin, which is in line with the highest level of liberated –NH_2_ (Table 1).

After entering the intestinal stage, the bands for the gastric digests of cooked skin disappeared rapidly after 15 min of intestinal digestion, and the remaining aggregates in the gastric digests of cooked triple and heart only appeared to be partially hydrolyzed. After the whole gastrointestinal digestion, obvious cardiac muscle fibers could still be observed in the digests of pork heart, whereas only smaller fibers could be found in the digest of tenderloin. This comparison further confirmed the lower digestibility of pork heart than tenderloin, as shown by the lower level of liberated –NH_2_ in Table 1. The structural different between cardiac muscle and skeletal muscle could account for their difference in digestibility. In addition, larger and smaller amorphous aggregates could still be observed in the digests of pork liver and triple, respectively. These images generally corresponded to the remained bands in the SDS-PAGEs of digested pork liver and triple (Figure 1). Notably, gastrointestinal digests of pork liver aggregated into aggregates bigger than 300 μm, thus corresponding to the lowest level of –NH_2_ in the digests of pork liver. The big aggregates in the digests of cooked liver and heart were difficult to be absorbed in the upper gut and will therefore be utilized by gut microbiota in the lower intestinal. Other than producing short-chain fatty acids (SCFAs), microbial protein metabolism can also result in additional potentially harmful compounds such as phenylpropionate, p-cresol, phenylacetate, indole propionate, indole acetate and amines [18,19,20]. Therefore, these partially digested proteins could be potential threats to the host. The smallest amount of undigested samples was found in the gastrointestinal digests of pork skin, despite the lower level of liberated –NH_2_ in this sample than those in the digests of tenderloin, pork heart and triple. It was speculated that skin was digested into relatively longer peptides due to the highest level of Pro and hydroxyproline residues, and some of these peptides could be further hydrolyzed by microvillus membrane hydrolases [21]. In addition, CLSM images indicate the affinity of lipid and protein, since the blue and red color are highly overlapped, indicating their close combination during digestion. Therefore, protein and lipid may affect the digestion process of each other. Fat was recently found to increase the digestibility of pork and chicken protein, possibly by changing the dispersion of protein during digestion [14]. Therefore, skin having the lowest digestibility might also be partly attributed to it having the lowest level of fat content.

### 3.3. Release of Free Amino Acids and Peptides

The levels of free amino acids in the gastrointestinal digest of each sample were compared in Table 2. Tenderloin released the highest content of free Ala, Glu, His, Lys, Tyr, carnosine and anserine, whereas heart digests contained the highest level of Leu, Met and Arg. Notably, tripe digests had the highest level of free Asp, Gln, Cys, Val, Phe, Pro, Ser, Thr, Ile and Asn, even though the liberated –NH_2_ in the digests of cooked tripe was much lower than that in cooked tenderloin and heart. The highest level of hydroxyproline was found in the digests of skin, which was in line with the high level of collagen in this sample. Free amino acids were generally of lower level in the digests of skin, corresponding to the lowest level of released –NH_2_ in the gastrointestinal digest of skin. Regarding the essential amino acid, tenderloin digests contained the highest amount of Lys and Trp, heart digests contained the highest amount of Leu and Met, even though Met was reported to inhibit the absorption of most amino acids [22]. Tripe digests contained the highest level of Val, Phe, Thr and Ile. Absorption of amino acids mixture is a complex competitive process, and therefore, the different level and composition determine the different nutritional value of cooked tenderloin and edible by-products to some degree [23].

The peptides in the gastric and gastrointestinal digests were analyzed to further uncover the digestibility profiles of these edible pork by-products. The total ions chromatograms of gastric or gastrointestinal digests were compared in Figure 3A–J. Considering that a hydrophobic column was applied in the separation of digests, gastrointestinal digests of liver and skin should contain more hydrophilic fractions, since more signals were observed in the initial time of the chromatograms, possibly due to differences in protein composition and digestion procedure between tested tenderloin and edible pork by-products.

Small peptides are more efficiently absorbed in small intestines than free amino acid mixtures, whereas the absorption of bigger peptides is uncertain [24]. The molecular weight distribution of peptides of each sample after gastric and the whole gastrointestinal digestion were compared in Figure 3K. After the simulated gastric digestion, higher percentage of peptides in the molecule range 400–1200 Da were identified in the digests of tenderloin and skin than others, corresponding to the higher level of liberated –NH_2_ in these two samples. After the whole gastrointestinal digestion, higher percentages of peptides in the molecule range >2000 Da were found in the digests of liver and skin, whereas higher percentages of smaller peptides (<1200 Da) were found in the digests of tenderloin and heart. These peptidomics results were also in line with the results of liberated –NH_2_, SDS-PAGE and CLSM images in previous sections, even though some small peptides were not considered due to the limitation of peptidomic algorithm [25]. Collagen is a major protein in skin. Some collagen is quite resistant to digestion because it has a complex triple helix structure and much higher percent of glycine and (hydroxyl) proline contributing to a tightly tertiary structure [26,27]. In addition, collagen has lower level of Trp, which are crucial action sites for both pepsin and chymotrypsin [28,29]. These account for the lower ratio of smaller peptides and larger ratio of bigger peptides in the digests of skin. Bioactive probabilities of the peptide with top 100 LFQ intensity in gastrointestinal digest were calculated, and the result was shown in Appendix A. Interestingly, the highest average bioactive probability was identified in the digests of liver (0.386), which was followed by the digests of tripe (0.385), tenderloin (0.336), skin (0.263) and heart (0.262).

In this work, all tested edible pork by-products were found to have lower digestibility than tenderloin, possibly due to the higher level of Pro, connective tissues, different protein composition or different muscle structures. High pressure, ultrasound treatment was reported to largely collapse the microstructures of muscle and connected tissues, which may help to increase the digestibility of these edible pork by-products [30,31]. Recent studies also indicated that high pressure, ultrasound treatment and mild salting help to improve the digestibility of myoglobin, which is of a rigid structure and is difficult to be digested [32,33,34]. Therefore, these methods can be applied in the processing of edible pork by-products to improve their digestibility and nutritional value in the meat industry.

## 4. Conclusions

The protein digestion process of selected edible pork by-products was concluded in Figure 4. Nearly half of digestion of skin protein occurred during the gastric stage, whereas intestinal digestion accounted for the major digestion of tenderloin and other edible pork by-products. The lowest gastric digestion occurred in cooked tripe, whereas relatively higher digestibility of tripe protein was found in the intestinal stage. Liver protein showed relatively lower digestibility during both gastric and intestinal stages, and heart protein showed relatively higher digestibility during both gastric and intestinal stages. Large-size aggregates still existed in the gastrointestinal digests of pork liver and heart. Digestion of pork tripe resulted in the highest level of free amino acids. These discrepancies in the digestion profile of tested edible pork by-products should be related to their different proteins compositions and muscle structures.

## Figures and Tables

**Figure 1 foods-11-03191-f001:**
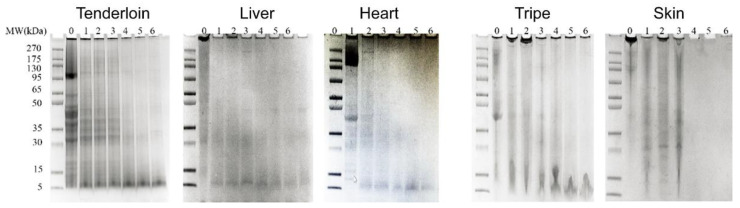
SDS-PAGE images of undigested and digested tenderloin and edible pork by-products. Lane 0 indicates undigested sample; lanes 1, 2 and 3 indicate samples after 15, 60 and 120 min of simulated gastric digestion, respectively; lanes 4, 5 and 6 indicate samples after 120 min of gastric digestion, which was followed by 15, 60 and 120 min of simulated intestinal digestion, respectively.

**Figure 2 foods-11-03191-f002:**
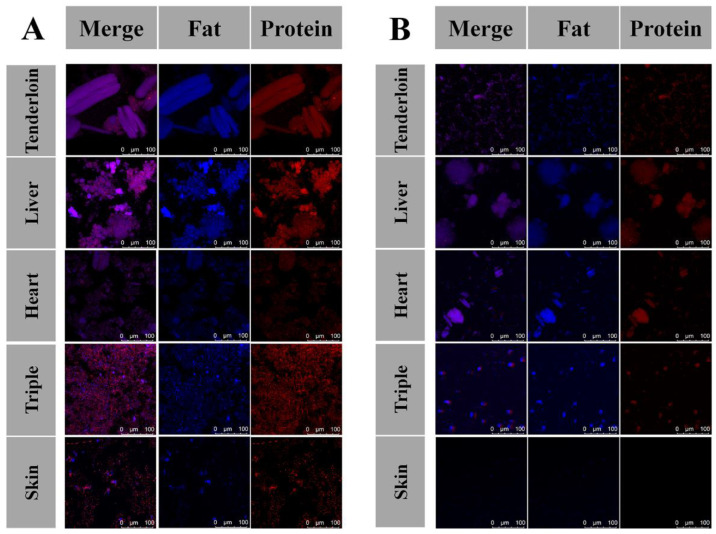
CLSM images of tenderloin and edible pork by-products after gastric digestion (**A**) and after the whole gastrointestinal digestion (**B**).

**Figure 3 foods-11-03191-f003:**
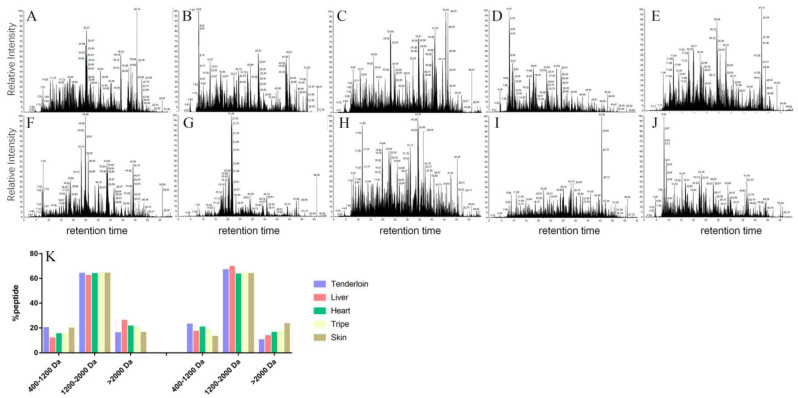
Comparison of the peptide compositions of tenderloin and edible pork by-product after the in vitro gastric and gastrointestinal digestion. (**A**–**E**) indicate the total ions chromatograms for the gastric digests of cooked pork tenderloin, liver, heart, tripe and skin, respectively; (**F**–**J**) indicate the total ions chromatograms for the gastrointestinal digests of cooked pork tenderloin, liver, heart, tripe and skin, respectively; (**K**) indicates the molecular distribution of gastric or gastrointestinal digests.

**Figure 4 foods-11-03191-f004:**
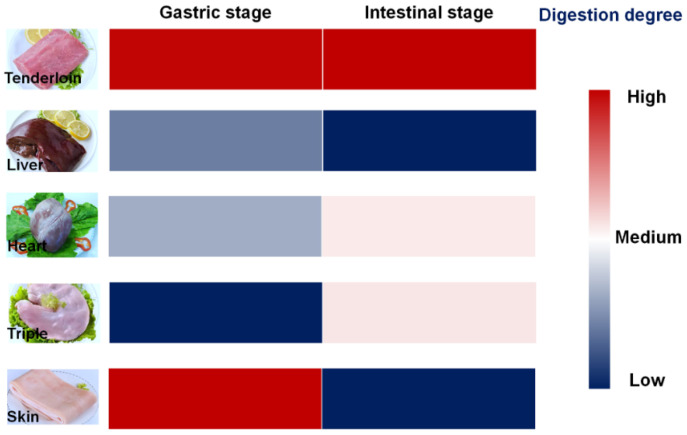
Digestion profiles of cooked edible pork by-products.

**Table 1 foods-11-03191-t001:** Liberation of primary amino group after the in vitro gastric (G_15_ and G_120_) and gastrointestinal (I_15_ and I_120_) digestion of tenderloin and edible pork by-product.

Sample ID	Concentration (μmol/g Protein)			
	G_15_	G_120_	I_15_	I_120_
Tenderloin	1.60 ± 0.02 ^B^	6.70 ± 0.33 ^A^	16.20 ± 2.99 ^A^	37.90 ± 3.71 ^A^
Liver	1.58 ± 0.80 ^B^	3.34 ± 0.20 ^B^	8.73 ± 0.23 ^C^	11.53 ± 0.38 ^E^
Heart	0.92 ± 0.08 ^C^	3.74 ± 0.10 ^B^	14.62 ± 0.16 ^B^	24.35 ± 2.11 ^B^
Tripe	1.01 ± 0.14 ^C^	1.90 ± 0.24 ^C^	7.35 ± 1.84 ^D^	21.31 ± 1.38 ^C^
Skin	3.28 ± 0.25 ^A^	6.84 ± 0.53 ^A^	7.40 ± 0.29 ^D^	14.94 ± 0.93 ^D^

The capital letters ^A–E^ indicate different significance (*p* < 0.05, n = 3) levels between the values within the same row (at the same digestion stage).

**Table 2 foods-11-03191-t002:** Comparison of liberated free amino acids of tenderloin and edible pork by-products after the gastrointestinal digestion.

AAs	Concentration of Free Amino Acids (μg/mL)				
	Tenderloin	Liver	Heart	Tripe	Skin
Ala	84.3 ± 6.5 ^A^	22.0 ± 3.4 ^B^	0.9 ± 0.2 ^E^	8.2 ± 1.5 ^D^	15.3 ± 2.4 ^C^
Glu	841.6 ± 53.7 ^A^	115.7 ± 9.3 ^D^	509.1 ± 45.1 ^B^	108.0 ± 8.4 ^D^	394.9 ± 41.0 ^C^
His	303.0 ± 26.9 ^A^	54.3 ± 4.8 ^B^	8.8 ± 1.1 ^C^	9.3 ± 0.4 ^C^	4.7 ± 0.6 ^C^
Lys	892.5 ± 78.5 ^A^	126.4 ± 11.7 ^C^	495.7 ± 53.6 ^B^	117.6 ± 10.7 ^C^	461.3 ± 52.7 ^B^
Trp	140.5 ± 16.1 ^A^	80.7 ± 5.2 ^C^	122.3 ± 10.5 ^B^	126.4 ± 13.9 ^B^	16.0 ± 2.4 ^D^
Tyr	63.1 ± 4.9 ^A^	16.4 ± 3.2 ^D^	51.4 ± 4.4 ^B^	42.5 ± 6.0 ^C^	13.9 ± 3.1 ^D^
Carnosine	398.9 ± 33.7 ^A^	11.5 ± 2.0 ^B^	7.7 ± 1.1 ^C^	1.4 ± 0.2 ^C^	ND
Anserine	21.9 ± 3.3	ND	ND	ND	ND
Leu	219.1 ± 18.6 ^D^	595.4 ± 49.5 ^C^	1129.7 ± 98.2 ^A^	934.1 ± 84.4 ^B^	138.5 ± 11.0 ^E^
Met	14.5 ± 1.8 ^C^	38.0 ± 3.5 ^B^	116.4 ± 14.0 ^A^	109.6 ± 9.3 ^A^	15.4 ± 2.6 ^C^
Arg	274.2 ± 21.7 ^B^	115.7 ± 12.8 ^C^	429.8 ± 42.7 ^A^	26.5 ± 2.9 ^D^	123.3 ± 14.1 ^C^
Asp	21.6 ± 1.4 ^C^	19.7 ± 2.9 ^C^	6.2 ± 0.7 ^D^	415.4 ± 37.2 ^A^	32.9 ± 2.5 ^B^
Gln	139.17 ± 10.4 ^B^	4.0 ± 0.4 ^D^	3.7 ± 0.4 ^D^	373.1 ± 37.4 ^A^	70.6 ± 4.8 ^C^
Cys	1.2 ± 0.1 ^C^	6.2 ± 0.5 ^B^	ND	41.8 ± 6.6 ^A^	7.2 ± 1.2 ^B^
Val	21.9 ± 2.7 ^C^	7.1 ± 0.6 ^D^	99.2 ± 8.4 ^B^	205.2 ± 22.0 ^A^	91.5 ± 5.1 ^B^
Phe	17.6 ± 2.0 ^D^	217.5 ± 17.2 ^B^	445.6 ± 36.1 ^A^	470.3 ± 42.8 ^A^	169.8 ± 14.5 ^C^
Pro	4.8 ± 0.7 ^D^	5.1 ± 0.8 ^D^	88.6 ± 5.3 ^B^	188.6 ± 19.3 ^A^	66.0 ± 4.9 ^C^
Ser	2.4 ± 0.4 ^D^	2.5 ± 0.7 ^D^	22.3 ± 1.3 ^B^	132.5 ± 14.2 ^A^	10.8 ± 2.0 ^C^
Thr	13.2 ± 1.9 ^D^	10.0 ± 1.4 ^D^	63.4 ± 7.6 ^B^	367.2 ± 31.7 ^A^	39.7 ± 5.1 ^C^
Ile	89.1 ± 5.8 ^D^	299.3 ± 24.2 ^C^	565.6 ± 47.3 ^B^	777.6 ± 68.4 ^A^	81.0 ± 6.5 ^D^
Asn	94.3 ± 10.8 ^B^	ND	7.2 ± 1.1 ^C^	334.2 ± 41.0 ^A^	ND
Hydroxyproline	7.4 ± 5.1 ^C^	1.1 ± 0.4 ^D^	1.3 ± 0.5 ^D^	21.7 ± 3.1 ^B^	157.9 ± 12.4 ^A^

ND is the abbreviation of not detected; The capital letters ^A^–^E^ indicate different significance levels (*p* < 0.05, n = 3).

## Data Availability

Data is contained within the article or Appendix A.

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
