# Peer review of "Digestion Profiles of Protein in Edible Pork By-Products"

_foods, 2022, doi:10.3390/foods11203191_

Round 1

Reviewer 1 Report

This work could be important, but it is a bit confusing in the presentation and discussion of the results. Most results are presented only, without any discussion (eg section 3.1.2. Release of free amino acids after complete gastrointestinal digestion). There are many laboritorial analyzes and results, and the discussion is very small. It presents an excess of Figures and some cannot be well understood and are useless. To be published, it would have to undergo a profound alteration, eliminating everything that is useless and improving the discussion.

The work must be published after several changes: How many samples were collected? Were they collected at the same time? How were they transported and preserved before analysis? The number of samples (n) must appear in the Tables.

No Quadro 1 há alguma confusão relativamente às diferenças significativas. No fim do quadro deve aparecer a seguinmte frase. “The capital letters A-E indicate different significance levels between values within the same line (p < 0.05).”

In Table 1 there is some confusion regarding the significant differences. At the end of the table the following sentence should appear. “The capital letters A-E indicate different levels of significance between the values within the same row (p < 0.05)”.

The section “Release of Free Amino Acids After Complete Gastrointestinal Digestion” (should be 3.1.3 and not 3.1.2) should be deleted. The results presented here are not discussed. They are never referred to at work again. In the same section 3.1.2., and in Figure 3, the colors of the graph are not clearly distinguished (eg ALA only the 1st color is different but all are significantly different; the same is true for Cys and Hydroxyproline). This entire point and the Figure must be eliminated. Figure 4 is very confusing. L, M, N and O charts can be eliminated. They are not needed. Table 2 What is it for? It is hardly mentioned in the discussion. It can also be deleted. In Conclusion, Figure 5 should be eliminated. After so many chemical analyses, the authors only refer the gastric and intestinal digestibility of the samples as a conclusion?

Thus, the work must be made major changes: Some data must be deleted and the discussion must be rewritten and reorganized.

Author Response

Dear reviewer

Thank you for the comments, which are helpful for the improvement of this work. The manuscript was revised based on the comments.

This work could be important, but it is a bit confusing in the presentation and discussion of the results. Most results are presented only, without any discussion (eg section 3.1.2. Release of free amino acids after complete gastrointestinal digestion). There are many laboritorial analyzes and results, and the discussion is very small. It presents an excess of Figures and some cannot be well understood and are useless. To be published, it would have to undergo a profound alteration, eliminating everything that is useless and improving the discussion.

Response: Thank you for the comments and suggestion. As suggested, the original Figure 4L-4O and Table 2 were deleted. The original Figure 3 was replaced by a Table (Table 2), in which the difference between samples can be illustrated more clearly. In addition, more discussions were made in the revised manuscript to improve the quality of this work (lines 234-235, 250-257, 268-270, 280-281, 310-318). Detail responses to the comments are as follows:

The work must be published after several changes: How many samples were collected? Were they collected at the same time? How were they transported and preserved before analysis? The number of samples (n) must appear in the Tables.

Response: Thank you for the suggestion. Three independent samples for tenderloin and each edible by-product were collected at the same time. They were preserved in box fill with ice bag and transported by car to a to freezer house (-20 °C) of our lab within 1 h after collection. These information was added in the revised manuscript (lines 66-68). The number of the sample was added in Table 1 and Table 2.

No Quadro 1 há alguma confusão relativamente às diferenças significativas. No fim do quadro deve aparecer a seguinmte frase. “The capital letters A-E indicate different significance levels between values within the same line (p < 0.05).”

In Table 1 there is some confusion regarding the significant differences. At the end of the table the following sentence should appear. “The capital letters A-E indicate different levels of significance between the values within the same row (p < 0.05)”.

Response: Thank you for the suggestion, we agree with this revision (line 172).

The section “Release of Free Amino Acids After Complete Gastrointestinal Digestion” (should be 3.1.3 and not 3.1.2) should be deleted. The results presented here are not discussed. They are never referred to at work again. In the same section 3.1.2., and in Figure 3, the colors of the graph are not clearly distinguished (eg ALA only the 1st color is different but all are significantly different; the same is true for Cys and Hydroxyproline). This entire point and the Figure must be eliminated. 

Response: Thank you for the suggestion. Since liberation of free amino acid are important results reflecting digestion profile of a protein, we decide to reserve the results but to show them using Table (Table 2), in which the difference between samples can be illustrated more clearly. In addition, we added more discussion regarding these results are added in the revised manuscript (lines 250-257).

Figure 4 is very confusing. L, M, N and O charts can be eliminated. They are not needed. Table 2 What is it for? It is hardly mentioned in the discussion. It can also be deleted. 

Response: Thank you for the suggestion, which are helpful for the improvement of this work. We agree with the viewpoint that Figure 4L-4O were deleted, and they were deleted in the revised manuscript. Table 2 was deleted in the main document and shown in the supplementary document.

In Conclusion, Figure 5 should be eliminated. After so many chemical analyses, the authors only refer the gastric and intestinal digestibility of the samples as a conclusion?

Response: Thank you for the suggestion. The conclusion was rewritten (lines 329-333). We decided to reserve this figure in this work, since readers can easily get the point of this work from this figure.

Reviewer 2 Report

The manuscript deals with the digestion profile of proteins from pork edible byproducts. Although lots of literature is available on the nutritive value of organ meat/ edible byproducts, the authors characterize the digestion profile of proteins which is a novel area of research. The research area and experimental design are novel. The manuscript is well written and has some significant findings. The language is clear and easy to understand. The hypothesis is well stated and clearly defined. I have the following comments as-

       i.          Please restrict the majority of discussion to the edible byproducts used in the present study such as liver, heart, tripe and skin rather than bones or feet/ trotters.

     ii.          The value in L 18 is bioactive probability?

   iii.          Line 26-28: Please cite suitable reference

   iv.          Line 32-34: seems irrelevant; may be given example of catgut, casing, collagen extraction etc.

     v.          L 45-46: please give examples of the byproducts evaluated in the present study only

   vi.          Line 54: transitional pork?

  vii.          L 142: please add further details about statistical analysis such as sample size, p value

viii.          Results and discussion: Appropriate

   ix.          Fig 5: Excellent concluding the experimental outcome.

Author Response

Dear reviewer

Thank you for the comments, which are helpful for the improvement of this work. The manuscript was revised based on the comments. In addition, the original Figure 4L-4O were removed, the original Table 2 was shown in the supplementary document, and the original Figure 3 were shown in the form of Table (Table 2), based on the comments of another reviewer. Detail responses to the comments are as follows:

The manuscript deals with the digestion profile of proteins from pork edible byproducts. Although lots of literature is available on the nutritive value of organ meat/ edible byproducts, the authors characterize the digestion profile of proteins which is a novel area of research. The research area and experimental design are novel. The manuscript is well written and has some significant findings. The language is clear and easy to understand. The hypothesis is well stated and clearly defined. I have the following comments as-

  1. Please restrict the majority of discussion to the edible byproducts used in the present study such as liver, heart, tripe and skin rather than bones or feet/ trotters.

Response: Thank you for the suggestion, we agree to restrict the majority of discussion to the edible byproducts used in the present study. Therefore, the introduction section was revised accordingly (lines 33-40). References were renewed based on these revision (reference 4 and 6).

  1. The value in L 18 is bioactive probability?

Response: Yes, the value here is the average bioactive probability of peptides in the digests (lines 18-19).

iii.          Line 26-28: Please cite suitable reference

Response: Thank you for the reminding. These value were calculated based on reference 1 and the number of slaughtered hogs in China in 2020. Therefore, reference 1 was cited here (line 29).

  1. Line 32-34: seems irrelevant; may be given example of catgut, casing, collagen extraction etc.

Response: Thank you for the comment. The content here was adjusted as suggested, example of catgut, casing and collagen were given here (lines 33-40).

  1. L 45-46: please give examples of the byproducts evaluated in the present study only

Response: Thank you for the suggestion. Examples including andouillette and Palatinate sausage were added here (lines 48-49).

  1. Line 54: transitional pork?

Response: It should be transitional pork here (line 59).

vii.          L 142: please add further details about statistical analysis such as sample size, p value

Response: Details about The sample size and p value were described in the revised manuscript (lines 150-153).

viii.          Results and discussion: Appropriate

Response: Thank you for the comment.

  1. Fig 5: Excellent concluding the experimental outcome.

Response: Thank you for the comment.

Round 2

Reviewer 1 Report

The article is much better, simpler and easier to read and understand.